Journal of
open psychology data

# Data from the Mixed Methods Project PICE (Parental Investment in Children's Education)

COLLECTION:
DATA FOR PSYCHOLOGICAL RESEARCH IN THE EDUCATIONAL FIELD

DATA PAPER

**MARIEKE HEERS** 

**SANDRA HUPKA-BRUNNER** 

**ANDRÉS GOMENSORO** 

**CHANTAL KAMM** 

*Author affiliations can be found in the back matter of this article

]u[ ubiquity press

## ABSTRACT

The Parental Investment in Children's Education (PICE) study focuses on Switzerland and investigates parental strategies, resources, and aspirations and how they shape their children's educational pathways. It contrasts families with a migration background to those without. PICE is a mixed-methods add-on-study of TREE (Transitions from Education to Employment). Within PICE one interview with young adults (N = 73, around age 20) and two interviews with one of their parents (N = 50) were conducted. The data are available for scientific analyses via SWISSUbase. They have reuse potential for analyses on parental investments, migration biographies as well as for methodological research on mixed methods.

**CORRESPONDING AUTHOR:**
**Marieke Heers**
FORS, CH
marieke.heers@unil.ch

**KEYWORDS:**
Education; migration; parental investment; mixed-methods; Switzerland

**TO CITE THIS ARTICLE:**
Heers, M., Hupka-Brunner, S., Gomensoro, A., & Kamm, C. (2023). Data from the Mixed Methods Project PICE (Parental Investment in Children's Education). *Journal of Open Psychology Data,* 11: 18, pp. 1–10. DOI: https://doi.org/10.5334/jopd.95

# (1) BACKGROUND

Across different contexts it has been observed that at comparable levels of parental socioeconomic status (SES) and educational attainment, children of migrants attain higher levels of education compared to the children of natives (Becker, 2010; Bodovski & Durham, 2010; Brinbaum & Kieffer, 2009; Glick & White, 2004; Kamm et al., 2022; Raleigh & Kao, 2010; Stevenson et al., 1990; Vallet, 1996). This has been explained by "immigrant optimism", which stipulates that migrant parents and their children express considerably higher educational aspirations than non-migrant families (Fernández-Reino, 2016; Kao & Tienda, 1995; Tjaden & Hunkler, 2017; Tjaden & Scharenberg, 2017).

Compared to their native counterparts, many migrant families (including those in Switzerland) have fewer resources, including socio-economic, cultural, social and linguistic capitals (Gomensoro & Bolzman, 2015; Pong et al., 2005; Stevenson et al., 1990; Wojtkiewicz & Donato, 1995). Their lower social origin and lack of resources makes migrant children generally less successful (perform worse) in the educational system than their native counterparts (Murdoch et al., 2017; OECD, 2012, 2015; Picot, 2012).

Studying parental aspirations is particularly interesting in Switzerland, where for a long time, the majority of migrants belonged to the lower socio-economic classes. Moreover, tracking into educational tracks happens early (around 12 years) and strongly determines further educational pathways. The Swiss education system is further characterized by a large proportion of youngsters who follow vocational training, which migrant parents are often less familiar with. At the same time, vocational education is highly valued in the labor market.

While an extensive literature has examined parental educational aspirations (e.g. Glick & White, 2004; Raleigh & Kao, 2010; Schnell et al., 2015; Spera et al., 2009), little is known about how they translate into specific behaviors and whether heterogeneities in parents' behavior explain why certain children experience successful educational trajectories while others do not. An important concept that sheds light on differential parental behaviors is that of parental investment, which seeks to analyze the intersection of parental aspirations, resources, and strategies of parents. A major reason for this research gap is that parental investment in children's education is hard to measure. A detailed description of PICE's theoretical framework is available in the technical report (Hupka-Brunner et al., 2022) and the scientific publications resulting from the project (e.g. Kamm et al., 2022; Kamm, Gomensoro, et al., 2023).

Against this background, the Parental Investment in Children's Education (PICE) project takes an interdisciplinary perspective and sets out to explore how the three dimensions of parental investment jointly contribute to children's educational success. More precisely, PICE examines the successful post-compulsory trajectories of children with parents of low levels of educational attainment and low SES, i.e. those who are considered as 'successful against the odds' (Rezai et al., 2015; Schnell et al., 2013; Schoon, 2006). Despite adverse conditions, children of migrant parents more often succeed against the odds; this observation has been made in different contexts (Brinbaum & Kieffer, 2009; Feliciano & Lanuza, 2017; Liu & White, 2017). The question of how parental investment helps children in succeeding against the odds is highly relevant in Switzerland, but also beyond. We emphasize that the aim of PICE is *not* to compare families in which children succeed against the odds with families in which they do not succeed, but to compare aspirations, resources, and strategies of successful families without any and with different migration backgrounds.

With its mixed-method approach, PICE aims at explaining parental strategies, resources and aspirations and how they shape their children's educational pathways (Hupka-Brunner et al., 2022). This is operationalized in three sub-questions: *(1) What is the effect of parental investments on educational success at the end of compulsory school? (2) What is the joint effect of parental investment and the educational situation at the end of compulsory education on post-compulsory outcomes (diplomas, educational pathways and satisfaction/ well-being)? (3) What are parental investment strategies and how are they adapted over time?*

On the one hand, PICE quantitatively studies the types of parental investments and their relationship with post-compulsory success. On the other hand, it qualitatively deepens the understanding of how parental aspirations and resources are mobilized within parental investment strategies. In the context of PICE, qualitative data has been collected. These data can be linked to the large quantitative study TREE2.

To do so, PICE takes advantage of the second cohort of TREE, Transitions from Education to Employment, TREE2 (Hupka-Brunner et al., 2021). TREE2 is a cohort of school-leavers who left compulsory school in 2016. PICE has added a qualitative part to TREE2, by drawing a sub-sample of respondents. The TREE2-respondents are interviewed once (four years after leaving compulsory school) and one of their parents twice (fourth and fifth year after finishing compulsory schooling). At the end of the first qualitative interview, the parents (N = 50) who participated in PICE, answered a short standardized questionnaire. With its interdisciplinary, longitudinal and mixed-methods setup, the PICE-data have reuse potential for a diversity of potential projects (see below). What makes the data also interesting for secondary analyses is that PICE has interviewed a cohort which entered the labor market or tertiary education at the time of a global pandemic.

This paper is based on the technical report of PICE (Hupka-Brunner et al., 2022). More details on its mixed methods design are also presented in a TREE Technical Paper (Kamm, Heers, et al., 2023). We advise interested readers to also consult those resources.

## (2) METHODS

### 2.1 STUDY DESIGN

To complement the quantitative TREE2 panel data (Hupka-Brunner et al., 2021), PICE has carried out a qualitative survey of a sub-sample of TREE2 participants. The PICE-data are a standalone dataset, i.e. they can be used for qualitative analyses independent of the TREE2-data and are made available separately. Upon special request and depending on the research question, both datasets can be used jointly for mixed methods analyses.

TREE2 is a follow-up survey of the Swiss large-scale Assessment on mathematics AES (Assessment of the Attainment of Educational Standards) which was carried out in 2016 by the Swiss Conference of Cantonal Ministers of Education. TREE2 covers a population of school leavers who left compulsory education in 2016 (n = 8.429).

In the context of PICE, young adults were interviewed at t4 in 2020. Their parents were interviewed twice, at t4 in 2020 and t5 in 2021. At the end of the t4 interview, parents also responded to a short questionnaire.

The methodological design and timeline of PICE are illustrated in Figure 1. The green bubbles represent the TREE2-survey-waves. 2016 was the last year of obligatory education and the end of lower secondary education.[1] Afterwards, the young adults started pursuing individualized educational pathways.

Interviewing young adults and their parents as an add on to a running panel study is an innovative design,

that allows studying parental investments and children's educational pathways in detail. At the same time, it was a challenge to insert PICE into the panel, organizationally as well as conceptually to not negatively impact the running panel study (for an in-depth discussion thereon see Hupka-Brunner et al. (2022) and Kamm et al. (2023)). We also emphasize that despite the elaborate design of the interviews, the results might be impacted by hindsight bias as parents were interviewed after their children had been successful.

### 2.2 TIME OF DATA COLLECTION

As shown in Figure 1, the qualitative interviews with TREE2-respondents and one of their parents took place in spring 2020 (following TREE2-wave t4). The start of data collection coincided with the first lockdown due to the Covid-19-pandemic. Therefore, the research team had to adjust the data collection that was initially planned as face-to-face interviews to remote interviews via video- and phone calls. In spring 2021, the second interview with parents was conducted. Due to the positive experience in 2020, the interviews in 2021 were also conducted remotely.

### 2.3 LOCATION OF DATA COLLECTION

PICE has been conducted in Switzerland. For several reasons, the focus is on the German- and French-speaking part of Switzerland, i.e. the Italian-speaking part has been excluded. In this respect PICE differs from TREE. The main reason was that in the Italian-speaking part, there are not enough cases of the different migrant cases in the TREE-data. Moreover, already with two language regions the project was extremely complex to handle; for example, in terms of interviewer recruitment and language competences required for data analysis.

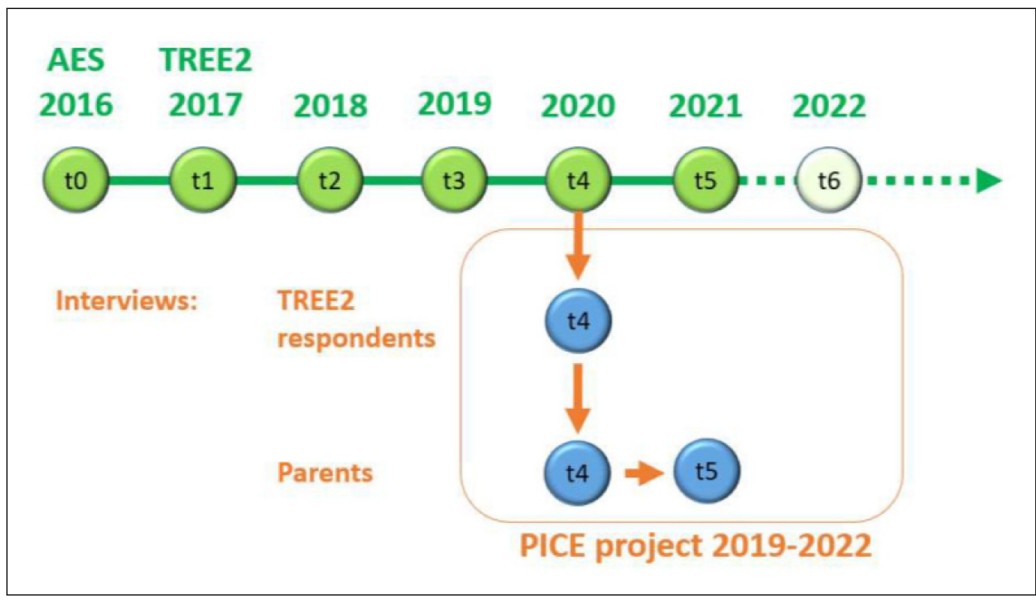

**Figure 1** Methodological design of PICE as an add-on study of TREE2.

 

## 2.4 SAMPLING, SAMPLE AND DATA COLLECTION

PICE-participants have been sampled based on the TREE2-data. A detailed overview on the variables that were used for the sampling procedure is provided as part of the documentation of the project and dataset (file *PICE_participants_TREE2variables.xls*).

Given the focus of the study, PICE-participants are individuals who succeed "against the odds". The conditions were defined as follows: (1) Respondents attend the pre-baccalaureate or extended requirement tracks at the end of compulsory education. (2) They have parents with modest social background (no tertiary education and low/medium socio-economic status (SES)). (3) Distribution by sex and German/French linguistic regions. (4) Distribution by country of origin, i.e. Switzerland, Southern EU (Portugal, Spain, Italy) or non-EU (former-Yugoslavia, Turkey, Sri-Lanka).

The aim was to include countries of origin that are well-enough represented in TREE2 and relevant for studying parental investments in the context of PICE.

Respondents who fulfilled these criteria were asked at the end of the TREE2-t4 survey if TREE could send them information on an in-depth-study (i.e. PICE). For PICE it was very useful to be able to draw from the panel loyalty of TREE-participants. Consent to participation was asked for before the interview.

Table 1 presents the characteristics of young adults and their parents who participated in the interviews. 73 TREE2-respondents were interviewed in 2020. 50 parents were interviewed in 2020 and 39 parents were again interviewed in 2021. The young adults were left with the choice of which parent should participate, according to which one is more involved in education. Table 1 shows the distribution across language regions, parents' country of origin and sex of the young adults. 72% (36 cases) of parents who participated in t4 were mothers, in t5 mothers again represented 72% of the cases (28 cases). For participation an incentive of 50CHF was paid.

## 2.5 MATERIALS/SURVEY INSTRUMENTS

A number of instruments and materials has been prepared and used for the collection of the qualitative data. These include: a contact form sent to respondents including a newsletter with FAQs, instructions for the interviewers, and the interview guide (all materials are provided via SWISSUbase; see below).

The interviews with young adults were conducted in German and French. Parents could choose whether they wanted to participate in German or French or in their mother-tongue. For this purpose, bilingual interviewers had been recruited (with mother tongue in the respective foreign language). If the interview was not conducted in German or French, it was translated to either French or German by the interviewer in the transcribing-process.

The content of the interviews was developed based on the research questions of PICE and to be complementary with TREE. This complementarity was crucial to analyze the research issues from different angles. For example, in the TREE-data, questions on "success" were measured objectively. In PICE they were measured subjectively to better understand what success means to respondents. Most topics were identical for the young adults and their parents. However, parents were asked more questions, for example, regarding their own educational and migration biography. Moreover, a short questionnaire was added. This included questions on biographic data, parents' professions, the number of children and some additional demographic information as well as questions on gender roles and two batteries on parental styles. For parental interviews in t5 for most parents, the focus was on changes and continuity so that the interviews could be kept rather short. The interviews included topics relating to educational events, aspirations, resources, and parental investment strategies as well as family life and migration biography.

For detailed information, see the interview guides and the technical report (Hupka-Brunner et al., 2022).

In TREE t4, a battery of questions has been added on parental strategies; it consists of seven items regarding parents' involvement with regard to TREE2 respondents' education at the end of secondary II (for the wording of the questions see documents PICE_added_questions_TREE2_t4 and the variable overview in the file PICE_participants_TREE2variables). The data will be made available with the TREE2-release of fourth survey wave.

| YOUNG ADULTS | GERMAN-SPEAKING REGION | | FRENCH-SPEAKING REGION | | TOTAL |
|---|---|---|---|---|---|
| | FEMALE | MALE | FEMALE | MALE | |
| Swiss | 4 (3/2) | 6 (6/4) | 7 (3/3) | 6 (5/5) | **23 (17/14)** |
| Southern European | 5 (3/1) | 6 (4/3) | 7 (6/5) | 5 (3/3) | **23 (16/12)** |
| Non-European | 10 (9/7) | 8 (5/4) | 6 (2/2) | 3 (1/0) | **27 (17/13)** |
| **Total** | **39 (30/21)** | | **34 (20/18)** | | **73 (50/39)** |

**Table 1** PICE qualitative participants in t4, parents between brackets (parents t5 in italics).

Source: Technical report PICE (Hupka-Brunner et al., 2022).

See also *PICE Material file* for an overview. These documents were constructed with the aim allowing for replication of similar studies.

## 2.6 QUALITY CONTROL

Collecting high-quality data that can be used for secondary analyses was a key objective of PICE. Throughout the project, several quality procedures were implemented into PICE. Particular attention was paid to the interview procedure and transcripts. The interview guidelines were developed based on well-established procedures and pre-tests were conducted. In addition to two days of training for the interviewers, the research team regularly verified the transcripts and, if needed, gave feedback and requested improvements regarding both interviewing and transcription.

All interviews were checked for quality. Two interviews with Swiss parents conducted in 2020 were very short and included fewer narrative passages than the other ones; these passages were important for the analysis. As there were sufficient Swiss cases, we decided not to interview them again in 2021 (t5). The two 2020-interviews are included the PICE data as they might be of interest for other researchers. The questions that were added to TREE2 at t4 as part of PICE were based on established indicators (see documentation).

When the data were deposited in SWISSUbase they were curated and checked for multiple quality criteria defined by international standards by data service experts.

## 2.7 DATA ANONYMISATION AND ETHICAL ISSUES

Prior to the interviews, informed consent was obtained; this has been recorded. For more details on these documents please consult the Technical Report PICE (Hupka-Brunner et al., 2022). The interview transcriptions have been de-identified. Given the mixed methods and longitudinal nature of the data, this was challenging. The aim was to preserve a maximum of information and, at the same time, protect the respondents and comply with data protection regulations. The anonymization implied the removal of proper names, cantonal educational programs, names of places and cantons. The details of the anonymization strategy can be found in the respective document (*anonymization-strategy*). Experts on data protection and research ethics have been involved in the development of the anonymization strategy. While the data cannot be fully anonymized, taken together with the user contract that must be signed to access the data (see 3.7) re-identification is prohibited. The approach chosen represents a well-reflected balance between data protection and keeping a reasonable amount of information. This is also why the PICE-data are made available without an identifier that allows linking them to the quantitative TREE2-data and to analyze them using mixed methods analyses. Upon special request access to the mixed methods data can be given (see 3.3).

## 2.8 EXISTING USE OF DATA

So far, the following publications have resulted from PICE:

Kamm, C., Gomensoro, A., Heers, M., & Hupka-Brunner, S. (2023). Parental Investment in Children's Educational Pathways: A Comparative View on Swiss and Migrant Families. *Swiss Journal of Sociology, 49*(2), 367–394. https://doi.org/10.2478/sjs-2023-0019

Kamm, C., Gomensoro, A., Heers, M., & Hupka-Brunner, S. (2022). Aspiring High in the Swiss VET-Dominated Education System: Second Generation Young Adults and Their Immigrant Parents. *Journal of Vocational Education & Training.* 1–20. doi:10.1080/13636820.2022.2139746

Kamm, C., Gomensoro, A., Heers , M. & Hupka-Brunner, S. (2021). Educational Aspirations of Migrant Parents and the Relationship With Educational Success. In C. Nägele, N. Kersh, & B. E. Stalder (Eds.), *Trends in vocational education and training research, Vol. IV. Proceedings of the European Conference on Educational Research* (ECER), Vocational Education and Training Network (VETNET) (pp. 119–129). https://doi.org/10.5281/zenodo.5180591

Kamm, C., Heers, M., Hupka-Brunner, S., & Gomensoro, A. (2023). Das Longitudinale Mixed Methods-Design der PICE-Studie (Parental Investment in Children's Education). *TREE Technical Paper Series, 2*, 1–15. https://www.tree.unibe.ch/unibe/portal/fak_wiso/c_dep_sowi/micro_tree/content/e206328/e305140/e305154/files1354202/Kamm_etal_2023_TREE_TP_2_ger.pdf

Hupka-Brunner, S., Heers, M., Gomensoro, A., Kamm, C. D. (2022). Parental Investment in Children's Education. A TREE2 mixed methods study. Technical Report. TREE / PICE. https://doi.org/10.48350/175906

Hupka-Brunner, S., Kamm, C., & Heers, M. (2022). Trotz allem erfolgreich – Bildungswünsche erfolgreicher Jugendlicher und ihrer Eltern aus schlechter gestellten Familien. *Transfer, Berufsbildung in Forschung und Praxis, 3*. SGAB, Schweizerische Gesellschaft für angewandte Berufsbildungsforschung. https://sgab-srfp.ch/trotz-allem-erfolgreich/

Hupka-Brunner, S., Kamm, C., Gomensoro, A., Heers, M. (2021). Berufswahl von Jugendlichen mit Migrationshintergrund: Wunsch und wahrgenommene Wirklichkeit. *terra cognita – Schweizer Zeitschrift zu Integration und Migration, Eidgenössische Migrationskommission EKM, 38*, 58–60. http://www.terra-cognita.ch/fileadmin/user_upload/terracognita/documents/terra_cognita_38_2021.pdf

Kamm, C., Gomensoro, A., Heers, M., & Hupka-Brunner, S. (2021). *Educational Aspirations of Migrant Parents and the Relationship With Educational Success.* Paper presented at the European Conference on Educational Research (ECER), Geneva (online). https://zenodo.org/record/5180591#.YbxPO2jMKUk

For an up-to-date overview, please consult the PICE-website.

## (3) DATASET DESCRIPTION AND ACCESS

### 3.1 REPOSITORY LOCATION
The data and documentations are located in SWISSUbase under the project number 20043 (https://www.swissubase.ch/en/catalogue/studies/20043/18606/overview) and have been assigned a permanent identifier: https://doi.org/10.48573/hjc1-3171.

### 3.2 OBJECT/FILE NAME
The PICE file set that can be accessed via SWISSUbase consists of three parts. (1) An overview of the material, (2) the technical report, and (3) the data and its documentation.

**(1)** The *Material Overview* provides an overview of the materials related to PICE. This includes the documentation as well as the datafiles. This file can be accessed via SWISSUbase without logging in and before making a request to download the data.

**(2)** The *Technical Report* (Hupka-Brunner et al., 2022) provides an in-depth description of the PICE-data and documentation, as well as the background of the study, including the theoretical framework, scientific motivations and research questions of PICE. It further describes the study design, data collection and processing and the structure of the data. The link to the TREE project is outlined and information on how to access the PICE data is provided.

There is considerable overlap between this paper and the technical report. We advise researchers who are interested in the PICE data to read the technical report first.

**(3)** The file set on SWISSUbase consists of the *data* themselves as well as the *documentation*. The documentation includes all the accompanying material as described below. The following data is included: interview transcripts in Word-format with parents at t4 and t5; interview transcripts in Word-format with the young adults at t4. The transcripts are available in either German or French, i.e. the langue in which the interview was conducted. If the interview was conducted in the native language of the parents, the translation into German or French. When downloading the data, researchers also download the documentation. This is subdivided into the instruments (interview guides for parents at t4 and t5 in German, French, Albanian, Italian, Portuguese, Serbian, Spanish, Turkish; and young adults at t4 in German or French). The closed questions asked to parents at t4 are provided in *.dta*-format (Stata) as well as csv-format. It further contains the communication material and description of the procedures that were used to contact the young adults and their parents. The communication folder also includes the consent form. The documentation further includes the material used for the interviewer trainings for the first and second field, both in German and French. As outlined above, PICE-participants are a sub-sample of TREE-respondents. An Excel summarizing the sample selection criteria is included in the documentation. As part of PICE some variables have been added to TREE. The wording and origin of the added variables are described in the documentation. As TREE is conducted in three languages (German, French, Italian) the wording of the added variables is outlined in the three languages. These data are not part of the PICE-dataset and will be part of TREE2 t4-data release. Finally, the documentation includes the anonymization strategy that has been developed for and implemented in PICE.

PICE data and documentation are downloaded in a *.zip*-file from SWISSUbase.

### 3.3 DATA TYPE
PICE-data are primary data. They consist of qualitative data, i.e. interviews with parents and young adults. Parents were interviewed at two time points, young adults at one. The PICE-data further contain quantitative data based on closed questions asked to parents at t4. While TREE and PICE are strongly connected, they are distributed separately. Mixed methods analyses are encouraged; yet, a specific request must be made via SWISSUbase to get access to the link.

### 3.4 FORMAT NAMES AND VERSIONS
The interview transcripts are available in Word-files (.docx). Parents' responses to the closed questions (t4) are available in a Stata-file (.dta) and csv-file. The documentation is mostly provided in PDF format, the file on sample selection in Excel (.xls).

### 3.5 LANGUAGE
The interview-data are provided in German or French, depending on the language in which they were collected.

In the cases where the parental interviews were conducted in a language different from German or French (i.e. Albanian, Italian, Portuguese, Serbian, Spanish, Turkish) they have been translated by the interviewers. The quantitative data on the closed questions obtained from parents are provided with variable names in English and variable labels mostly in German. TREE is conducted in three languages (German, French, Italian) the wording of the added variables in t4 as part of PICE was also added in the three languages.

For each file, the *Material Overview* indicates the language.

### 3.6 LICENSE
The PICE data are available in SWISSUbase under the standard contract for social sciences "Restricted Contract": https://www.swissubase.ch/en/catalogue/studies/20043/18606/datasets/2141/2523/contract/usage-license.

### 3.7 LIMITS TO SHARING
The PICE-data contain sensitive information. In compliance with data protection regulations (Swiss Federal Law), the datasets are available for scientific research and academic teaching only. Data can be obtained and processed by researchers after signing a data use agreement. For that purpose, an account in SWISSUbase and an affiliation with an academic institution are required. Researchers must also submit a short description of the planned research project. For details on the PICE-user contract see here: https://www.swissubase.ch/en/catalogue/studies/20043/18606/datasets/2141/2523/contract/usage-license.

Documentation such as the technical report is accessible without restrictions from the website.

### 3.8 PUBLICATION DATE
The first version of the PICE-data and -documentation were published on 19/01/2023.

### 3.9 FAIR DATA/CODEBOOK
SWISSUbase conforms to the FAIR principles *(Findable, Accessible, Interoperable, Reusable)* and is recognized by the Swiss National Science Foundation. *Findability* of the data is ensured by a unique and persistent identifier (DOI). Rich social science-discipline-specific metadata are provided and freely accessible via SWISSUbase. The data can be found by searching the SWISSUbase public catalogue which is accessible. The data are also findable via the CESSDA catalogue. *Accessibility* of the PICE-data is possible with user authentication and under the conditions outlined in section 3.7 and here. Accessibility is further assured by the fact that SWISSUbase is multilingual and ensures long-term preservation of the data. *Interoperability* is given based on the metadata profile used in SWISSUbase and the controlled

vocabularies based on discipline-specific international standards (DDI in this case). Finally, *Reuse* of the data is possible under the conditions outlined in section 3.7. The PICE-data are accompanied by rich discipline-specific metadata and documentation that allow secondary users of the data to understand and analyze them, as the aim of sharing data is to allow for similar studies in other contexts. Additionally, quality assurance checks have been carried out by data service experts to ensure coherence of data, metadata and documentation. To access the SWISSUbase platform, visit: www.swissubase.ch. Click here for general information about the SWISSUbase platform and services.

## (4) REUSE POTENTIAL

From the start, PICE was designed in a way to allow secondary analyses and we strongly encourage researchers to do so. Preparing the data for sharing and reuse was also encouraged and financially supported by the funder of the study, the Swiss National Science Foundation (SNSF). The qualitative data can either be analysed separately or PICE can be analysed by mixed-methods analysis jointly with the TREE data (upon request as specified in 3.3). Providing in-depth and high-quality documentation was an objective of our project. The ambition was to create a project that can also be used as a model of how to present and document qualitative and mixed-methods research.

Given the interdisciplinary nature of PICE it can be used for analyses from different disciplines, including educational sciences, migration studies, psychology, and life course research.

Moreover, the context of the data collection during the first lockdown of the COVID19-pandemic in spring 2020 (see section 2.2) allows for analyses on the impact of the pandemic on educational trajectories and decisions.

While PICE has led into several publications, there are many remaining opportunities the data could be used for. For example, the change of parental aspirations between the two waves of data collection could be analysed. Another interesting topic would be to study the discrepancy between reports of parental aspirations by the young adults in the quantitative data and parents' own reports in the qualitative data. Gender comparisons and research questions on intersectionality are also interesting and highly relevant topics for further analyses of the PICE-data. The data are also very suitable for doctoral theses.

So far, in Switzerland, PICE is one of the largest mixed methods and qualitative studies for which the data are made available for secondary analysis.

The PICE data were collected in an educational system where VET is common and highly valued in the labour market. Findings based on PICE are probably to some

extent generalizable to contexts with similar educational systems, such as Germany. It could be interesting to conduct comparative analyses. Yet, it must be kept in mind that the migration history and current context is different from other European countries.

While the PICE-data offer numerous opportunities for secondary analyses, given the qualitative nature of the data the generalizability of the findings is limited.

Whether researchers work only with PICE or in a mixed methods framework depends on the research question. As described above, PICE allows for numerous qualitative analyses related to parental investment in families where children succeed against the odds. If, on the other hand, researchers aim at analyzing the full population of young adults or pathways, it would be more adequate to work with TREE. We advise potential users of the data to consult the technical reports of both and then ask for access to the dataset that best fits their research needs. We advise researchers first to see if either PICE or TREE is adequate for addressing their research question(s). If the question requires both data sources the matched data can be requested.

Finally, we emphasize that the PICE-data are also a rich resource for teaching. Either methodological or substantial courses on education and migration. In combination with the TREE-data they can further be used for teaching mixed methods.

Depending on research interest, the PICE-research team might be available for cooperations.

## NOTE

1  For an overview of the Swiss educational system please consult the website of the Swiss Conference of Cantonal Ministers of Education https://www.edk.ch/en/education-system/diagram?set_language=en.

## SPECIAL COLLECTION

This submission is part of the Special Issue "Data for Psychological Research in the Educational Field" edited by Sonja Bayer, Katarina Blask, Timo Gnambs, Malte Jansen, Débora Maehler, Alexia Meyermann and Claudia Neuendorf (alphabetic order).

## ACKNOWLEDGEMENTS

We are very grateful to the TREE-team at the University of Bern to provide us with the opportunity to carry out PICE as an add-on study and for their valuable support throughout the project. We further thank the Data Archive Service and Data Management Support team at FORS, the Swiss Centre of Expertise in the Social Sciences, for their support regarding data archiving, management, and research ethics. We further thank Jacob Schnell and Ana Texeira as well as the interviewers for their excellent work. Finally, we thank the project partners Laura Bernardi and Ben Jann and the PICE advisory board, Christophe Delay, Andreas Hadjar, Katariina Salmela-Aro and Eva Mey, for their valuable feedback and support.

## FUNDING INFORMATION

The PICE-project was funded from 01.08.2019 to 31.01.2023 by the Swiss National Science Foundation (SNSF) under the grant-number 100019_184906.

## COMPETING INTERESTS

The authors have no competing interests to declare.

## AUTHOR AFFILIATIONS

**Marieke Heers** [iD] orcid.org/0000-0002-0479-910X
FORS, CH

**Sandra Hupka-Brunner** [iD] orcid.org/0000-0002-8538-4079
University of Bern, CH

**Andrés Gomensoro** [iD] orcid.org/0000-0003-3969-2414
University of Bern, CH

**Chantal Kamm** [iD] orcid.org/0000-0001-8568-5659
Bern University of Teacher Education, CH

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

## PEER REVIEW COMMENTS

*Journal of Open Psychology Data* has blind peer review, which is unblinded upon article acceptance. The editorial history of this article can be downloaded here:

- **PR File 1.** Peer Review History. DOI: https://doi.org/10.5334/jopd.95.pr1

**TO CITE THIS ARTICLE:**
Heers, M., Hupka-Brunner, S., Gomensoro, A., & Kamm, C. (2023). Data from the Mixed Methods Project PICE (Parental Investment in Children's Education). *Journal of Open Psychology Data,* 11: 18, pp. 1–10. DOI: https://doi.org/10.5334/jopd.95

**Submitted:** 31 May 2023    **Accepted:** 26 October 2023    **Published:** 27 December 2023

*Journal of Open Psychology Data* is a peer-reviewed open access journal published by Ubiquity Press.

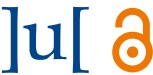