## [Peer Review History. · Journal of Open Psychology Data]

For: Journal of Open Psychology Data

Title: Data from the mixed method project PICE (Parental Investment in Children's Education)

September 13, 2023

Dear Reviewers,

We have received your comments on our paper, submitted to the Journal of Open Psychology Data. First, let us express our gratitude for your constructive remarks. They helped us a lot in revising and improving the paper. We believe that we were able to adequately address most of your comments. We hope that you will be satisfied by the revisions and our replies.

Below is a point-by-point response to your comments.

Yours faithfully,

The authors

Reviewer 1:
Recommendation: Revisions Required

Comments to the author(s)

I had the opportunity to review the submission "Data from the mixed method project PICE (Parental Investment in Children's Education)". Overall, I found the submission to be clear and complete. I have some questions, however, and some minor issues that could improve readability.

Abstract, last sentence: I suggest omitting "considerable" from this sentence, such evaluations depend very much on researchers objectives.

- We have omitted this word from the abstract and also from the background section.

Background:

I found some redundancies in the first paragraph, particularly to last sentence contains novel references but the statement has been made before in that paragraph. I suggest revising the paragraph.

- The paragraph has been revised, in particular, the last sentence has been merged with the first part of the paragraph.

From a German perspective, the VET system is quite common. What makes the data interesting for researchers from countries with a similar educational system?

- We have added the following to the Reuse potential section:
 - o "The PICE data were collected in an educational system where VET is common and highly valued in the labour market. Findings based on PICE are probably to some extent generalizable to contexts with similar educational systems, such as Germany. It could be interesting to conduct comparative analyses. Yet, it must be kept in mind that the migration history and current context is different from other European countries."

Please provide references for the statement "While an extensive literature has examined parental educational aspirations".

- We have added the following: "(e.g. Glick & White, 2004; Raleigh & Kao, 2010; Schnell et al., 2015; Spera et al., 2009)

"...strategies of parents in interaction" => interactions with whom?

- This sentence was indeed not clear. It has been reformulated into "analyze the intersection of parental aspirations, resources, and strategies of parents".

How does the concept of parental involvement in school relate to the conceptualization of aspirations, resources, and strategies? Furthermore, aspirations, resources, and strategies seem to have specific interrelations as well, maybe in terms of a mediation, more details on these concepts would be needed to understand the theoretical framework of the data.

- In PICE, parental involvement has been defined as being composed of parental aspirations, resources and strategies. As the theoretical framework has been described at several places, for this data paper, we decided not to repeat all the details. Instead, in the revised version, we now refer more explicitly to the other resources where readers can find a detailed description of the theoretical framework. We now write that "A detailed description of PICE's theoretical

framework is available in the technical report (Hupka-Brunner et al., 2022) and the scientific publications resulting from the project (e.g. Kamm et al., 2022; Kamm, Gomensoro, et al., 2023)."

"successful post-compulsory trajectories of children with parents of low levels of educational attainment and low SES" => are comparison groups considered (if not, why?), e.g., disadvantaged and not successful, or advantaged and not successful? That would be necessary to evaluate whether findings are unique for the "success against the odds", or whether these parental strategies are merely widely used.

- The focus of PICE was on families that "succeed against the odds". These families were the population of interest. It was not the aim to compare succeeding and non-succeeding families but to compare successful families with different origins (Swiss vs. migrant families) to see whether parental investment in these families differs.
- We also emphasize that in PICE, we focus on variance in commonality (Glaser & Strauss, 2010¹). The relevant variance criteria of the research question were taken into account, while holding the rest of the contextual conditions as similar as possible.
- Hence, based on PICE no conclusions on non-successful families can be drawn. Comparisons between successful and non-successful families would be interesting and relevant but are beyond the scope of this project. Other projects should be designed in a way to allow for comparison.
- At the end of the background section, we now refer to this and write: "We emphasize that the aim of PICE is not to compare families in which children succeed against the odds with families in which they do not succeed but to compare families without any and with different migration backgrounds."

Resources and aspirations seem to be constructs that researchers could better measure in a standardized way, is there quantitative data on these as well?

- In TREE these constructs are measured and it is indeed one of the aims of PICE to assess them both quantitatively and qualitatively and to contrast the qualitative and the quantitative results. PICE has been valuable in this respect as it revealed that with both data sources we capture different nuances of parental aspirations and families' resources. The mixed methods approach lead to new insights compared to the existing literature. For example, we could show that Swiss parents more often say that they have no specific aspirations for their (successful) children. At first, this seemed counterintuitive, because the literature has shown that clear and high aspirations are often a driver of educational success. With the qualitative data we could show that "no aspirations" stands for an open and trustful perspective of parents that want their children to be happy, to work in a domain that really matches their children's needs and competences.

From my perspective, qualitative approaches are not adequate to investigate "effects" – what do the researchers mean with "effects"?

- We agree that the wording might be unclear in relation to qualitative data. In fact, the research questions in which we refer to "effects" stem from the formulation of the research questions in initial research proposal submitted to the funder and we have decided to not change the wording in the paper. The idea was to analyze this part of the proposal with the quantitative data TREE data.

¹ Glaser, B. G., & Strauss, A. L. (2010). *Grounded theory: Strategien qualitativer Forschung* (3rd ed.). Huber.

- In the other sentence in the background section, we have reworded "effects" into "relationship with".

In the paragraph starting with "On the one hand, PICE quantitatively...", a brief overview of the types of data should be provided to understand that PICE has quantitative and qualitative data and is linked to the large quantitative study TREE2. This was difficult to understand and only became clear later.

- This has been clarified. We have added the following sentence to the paragraph: "In the context of PICE, qualitative data has been collected. These data can be linked to the large quantitative study TREE2." Moreover, as outlined in the paper, in the context of PICE, we have developed a short questionnaire for parents.

Quantitative questionnaires => standardized questionnaires or questionnaires with standardized measures, the questionnaires themselves are not quantitative

- We have reformulated this in "standardized questionnaire".

The researchers promote their data, which is good, but they should keep in mind the limited generalizability that goes along with qualitative data, even though the theoretical sampling is a great strength here.

- Thanks for pointing this out. To overcome the shortcomings of both qualitative and quantitative data PICE has taken a mixed-methods lens and allows linking the qualitative data to the cohort data from TREE.
- We have added a sentence on this to the section on the reuse potential: "While the PICE-data offer numerous opportunities for secondary analyses, given the qualitative nature of the data the generalizability of the resulting findings is limited."
- Yet, we want to emphasize that from a data management perspective, it becomes more common and more relevant to archive qualitative data and to make them available for secondary analyses. This is also the case for the platform SWISSUbase. There is a growing understanding that there is large reuse potential of qualitative data, for research and teaching.

Another aspect brought forward in section 2.1 is the potential hindsight bias of parents because they were interviewed after their children had been successful.

- We have added this to the text. "We also emphasize that despite the elaborate design of the interviews, the results might be impacted by hindsight bias as parents were interviewed after their children had been successful."
- We assume that the parental strategies reported at the moment of the interviews can be interpreted in relation to the success of the children. Parental behaviors are defined by the success of their children, i.e. they would behave differently if their children were not successful.
- Another way to look at the data is to consider parental support and strategies as their narratives and to analyze if these narratives changed between the two interview waves.

In Figure 1, please indicate when respondents have left secondary school.

- Depending on the educational pathways this differs. Therefore, we have decided not to integrate this information into the figure. We have added to the text that "2016 was the last year of obligatory education and the end of lower secondary education. Afterwards, the young adults started pursuing individualized educational pathways". Moreover, we have added a reference to the Swiss educational system in a footnote:

"For an overview of the Swiss educational system please consult the website of the Swiss Conference of Cantonal Ministers of Education https://www.edk.ch/en/education-system/diagram?set_language=en".

Section 3.2, paragraph (2)

...provides "an in-depth" (not "and in depth")

- This has been corrected.

Structure "of the data" (not "to the data")

- This has been corrected.

"This article is very much..." => sentence is redundant

- The sentence has been deleted.

Section 3.2, paragraph (3)

The word "documentation" may lead to misunderstandings because the technical report is documentation as well, may "accompanying material" or "supporting material" is a better choice of words.

- This is due to the structure and naming of files in SWISSUbase. We agree that it might be misleading. It is not possible to change that in SWISSUbase (at least not in the short term). Therefore, in the manuscript, we have reformulated and added more precision to the text: "The file set on SWISSUbase consists of the data themselves as well as the documentation. The documentation includes all the accompanying material as described below."

Please provide data in accessible format, Stata format belong to a proprietary software, csv would be accessible.

- Thanks a lot for pointing this out. The same concern has been raised by Reviewer 2 as well. We have created a CSV-version of the data that will be made available in addition to the Stata-file. This implies that a new version of the data must be made available and a new doi is provided. Moreover, the data needs to be curated again. This will take a couple of weeks. We have agreed with the reviewer that the curation take place in parallel to the second round of the review.

3.3, last sentence contains "must be made" twice

- This has been corrected.

3.5, "provided with variable names in English and variable names mostly in German" => is one "variable names" something else?

- There was a mistake in the initial text, it should be "variable labels mostly in German". This has been corrected.

(4), last sentence: "they can further used for teaching mixed methods" => "they can be used for teaching mixed methods"

- This has been corrected.

Reviewer 2:
Recommendation: Revisions Required

Comments to the author(s)

Thank you for the opportunity to read this interesting manuscript! In the present data publication, the authors introduce a mixed methods dataset on the parental investment of migrants in Switzerland. I fully agree with the authors that this is indeed a highly relevant topic and a dataset that has considerable reuse potential for future research on this topic. The manuscript is overall well-written and the data is comprehensively documented in various sources. However, I have unfortunately not been able to access the deposited data which is why I am not entirely sure whether or not they fully comply with the open access guidelines of the Journal of Open Psychology Data.

- This has been clarified with the editors prior to submitting the first version of the manuscript. We also emphasize that this how SWISSUbase treats most datasets that include sensitive information. The general rule is "as open as possible, as closed as necessary". This is also in line with the mantra of the Journal of Open Psychology Data ². Therefore, we speak of *accessible* rather than *open* data.
- Completely open data is not in line with data protection requirements and consent, even though we have anonymized the data.

Deposited data

- The data must be deposited under an open license that permits unrestricted access (e.g. CC0, CC-BY). Data access is currently limited as outlined in sections 3.6 and 3.7 of the manuscript. In my personal opinion, the license model that is described in these sections is acceptable due to the sensitive nature of the data. However, in the end it is the editor's call to decide whether the specific license is acceptable for JOPD.

- The deposited data must include a version that is in an open, non-proprietary format.

Data is currently available in Stata format (dta) only (at least this is stated in the description but as said above I cannot access the data itself). The authors might simply consider to provide an additional version of their data in a non-proprietary format (csv etc.) to address this issue.

- Thanks a lot for pointing this out. The same concern has been raised by Reviewer 1 as well. We have created a CSV-version of the data that will be made available in addition to the Stata-file. This implies that a new version of the data must be made available and a new doi is provided. Moreover, the data needs to be curated again. This will take a couple of weeks. We have agreed with the reviewer that the curation take place in parallel to the second round of the review.

Remarks on the manuscript:

The manuscript summarizes all important information on the data and how it was collected in a concise manner. I have only a few comments on the manuscript:

Connection to TREE data

The authors describe how the PICE data and the TREE data are connected in a detailed manner at the start of the methods section. Readers might consider how to combine these data to answer their

² <https://openpsychologydata.metajnl.com/about>

research questions at this point. Thus, it is slightly disappointing to learn in section 2.7. that “PICE-data are made available without an identifier that allows linking them to the quantitative TREE2-data and to analyze them using mixed methods analyses”. Maybe this could be addressed by transparently reflecting this issue earlier in the methods section and mentioning that the presented PICE dataset is a standalone dataset rather than a complement (e.g. in section 2.1). Just to clarify, I fully understand that there are valid ethical concerns here and that there are good reasons for proceeding in this way, I just would like to have this issue to be reflected more transparently at an earlier point of the manuscript. For example, including the following paragraph on TREE t4 data in the survey instruments section might result in the impression that TREE and PICE data can be matched: “In TREE t4, a battery of questions has been added on parental strategies; it consists of seven items regarding parents’ involvement with regard to TREE2 respondents’ education at the end of secondary II (for the wording of the questions see documents PICE_added_questions_TREE2_t4 and the variable overview in the file PICE_participants_TREE2variables). The data will be made available with the TREE2-release of fourth survey wave.”

- We agree that it would be useful to add more transparency on this issue. We have added the following text to section 2.1:
 - o “The PICE-data are a standalone dataset, i.e. they can be used for qualitative analyses independent of the TREE2-data and are made available separately. Upon special request and depending on the research question, both datasets can be used jointly for mixed methods analyses.”

Finally, I would suggest to provide in the “Reuse potential” section some information on the circumstances under which it makes sense for users of the PICE data to also access TREE data without a special request (and maybe also more information on what exactly “a specific request must be made via SWISSUbase” means). Are there potential reuse scenarios for the “standard datasets” of TREE and PICE without this specific request even though the data cannot be matched or would the authors always advise users to request to mixed methods dataset?

- Whether researchers want to only work with PICE or in a mixed methods framework depends on the research question. As described above, PICE allows for numerous qualitative analyses related to parental investment in families where children succeed against the odds. The focus of PICE is on migrant families. If, on the other hand, researchers aim at analyzing the full population of young adults or pathways, it would be more adequate to work with TREE.
- We advise potential users of the data to consult the technical reports of both and then ask for access to the dataset that best fits their research needs.
- TREE data is also available upon prior agreement, i.e. researchers must indicate their research question and analysis aim.
- We would not advise to always request the mixed methods data. We suggest to see if either PICE or TREE is adequate for addressing their research question(s). If the question requires both data sources we advise requesting the matched data.
- We have added the following to the reuse section: “Whether researchers work only with PICE or in a mixed methods framework depends on the research question. As described above, PICE allows for numerous qualitative analyses related to parental investment in families where children succeed against the odds. If, on the other hand, researchers aim at analyzing the full population of young adults or pathways, it would be more adequate to work with TREE. We advise potential users of the data to consult the technical reports of both and then ask for access to the dataset that best fits their research needs. We advise researchers first to see if either PICE

or TREE is adequate for addressing their research question(s). If the question requires both data sources the matched data can be requested.”

- A specific request implies that researchers outline in SWISSUbase what they plan to do with the data. This is then sent to the data producers who take a decision if access is given. This is outlined in section 3.7. We emphasize that we favor mixed-methods analyses and want to give access to the mixed-methods data whenever the research question and data protection requirements allow it.

Section 3.9.

I think it is sufficient to mention that a DOI is provided as a persistent identifier once in this section.

- The second reference to the DOI has been deleted from that section.

References

Hupka-Brunner, S., Heers, M., Gomensoro, A., & Kamm, C. D. (2022). Parental Investment in Children's Education. A TREE2 mixed methods study. Technical Report. TREE / PICE.

I might be convenient to include the corresponding URL here (I am aware that it is mentioned in a footnote but I would also include it in the corresponding reference).

- We have added the doi to the reference.
- Hupka-Brunner, S., Heers, M., Gomensoro, A., & Kamm, C. D. (2022). *Parental Investment in Children's Education. A TREE2 mixed methods study*. Technical Report. <https://doi.org/10.48350/175906>

Typos:

in-dept -> in-depth

- This has been corrected.

„in“ is crossed out in section 3.9

- This has been corrected.